# Drone-based displacement measurement of infrastructures utilizing phase information

Shien Ri [1] ✉, Jiaxing Ye [1] ✉, Nobuyuki Toyama [1] & Norihiko Ogura[2,3]

Drone-based inspections provide an efficient and flexible approach to assessing aging infrastructures while prioritizing safety. Here, we present a pioneering framework that employs drone cameras for high-precision displacement measurement and achieves sub-millimeter accuracy, meeting the requirements for on-site inspections. Inspired by the principles of human auditory equilibrium, we have developed an effective scheme using a group of strategical reference markers on the bridge girders to measure structural displacements in the bridge. Our approach integrates the phase-based sampling moiré technique with four degrees-of-freedom geometric modeling to accurately delineate the desired bridge displacements from camera motion-induced displacements. The proposed scheme demonstrates favorable precision with accuracy reaching up to 1/100th of a pixel. Real-world validations further confirmed the reliability and efficiency of this technique, making it a practical tool for bridge displacement measurement. Beyond its current applications, this methodology holds promise as a foundational element in shaping the landscape of future autonomous infrastructure inspection systems.

Civil infrastructure, encompassing indispensable physical systems and edifices such as thoroughfares, overpasses, and communal amenities, is vital for facilitating economic expansion, fostering societal unity, and enhancing overall living standards. However, the issue of aging infrastructure has emerged in recent decades as a major challenge facing contemporary society. This challenge is particularly pronounced in the case of bridges, which constitute the linchpin of contemporary civilization. According to official sources[1,2], the proportions of bridges over 50 years old in the United States and Japan currently stand at 42% and 43%, respectively. Additionally, the European Commission reports that approximately 10% of bridges are over 100 years old[3]. Furthermore, it is essential to acknowledge another crucial statistic concerning deficient bridges that are enduring deteriorated conditions due to heavy usage or hostile environment, posing a significant risk to society. According to America's Infrastructure Report Card (ASCE) 2021, 7.5% of all bridges in the United States are classified

as structurally deficient. Regrettably, despite this classification, an alarming 178 million trips are still made daily across these deficient bridges. Similarly, in Japan, the 2023 official annual report of infrastructure maintenance[4] revealed that over 10% of bridges were categorized as stage III deterioration conditions, indicating the urgent need for early rehabilitation and maintenance to ensure the structural integrity of these bridges for continued service. The aging and deficient infrastructure can have severe consequences for society in various aspects, such that it poses a safety risk to the inhabitants, requires significant time for repair or replacement, causes economic impacts due to out-of-service conditions, and gives rise to broader social concerns. In light of the numerous detrimental consequences, the Organization for Economic Co-operation and Development recommends that nations invest in the modernization of their aging infrastructure and endeavor to allocate an average of 3.5% of their annual Gross Domestic Product towards infrastructure development[5]. The

[1]Research Institute for Measurement and Analytical Instrumentation, National Institute of Advanced Industrial Science and Technology (AIST), Central 2, 1-1-1 Umezono, Tsukuba, Ibaraki 305-8568, Japan. [2]CORE Institute of Technology Corporation, 3-8-5 Asakusabashi, Taitou-ku, Tokyo 111-0053, Japan. [3]iTi Laboratory, Department of Civil and Earth Resources Engineering, Graduate School of Engineering, Kyoto University, Goryoohara, Nishikyo-Ku, Kyoto 615-8245, Japan. ✉e-mail: ri-shien@aist.go.jp; jiaxing.you@aist.go.jp

costs of infrastructure maintenance and replacement are vast, but the costs of failing to conduct investment properly in infrastructure management are incalculable.

To guarantee the sustainability of civil structures, it is imperative to utilize efficacious and cost-effective non-destructive evaluation methods that can assess the integrity and condition of structures without causing damage, making them valuable tools for maintaining and extending the life of infrastructure assets[6]. Bridges can be subjected to various loads such as traffic, wind, and temperature changes, which can cause deflection, i.e., vertical in-plane displacement. Deflection measurement constitutes a crucial component in evaluating the condition of civil structures, as it furnishes indispensable insights into a bridge's structural integrity and further facilitate engineers identify potential structural problems and ensure the safety and longevity of the bridge. Nevertheless, conventional sensing devices, such as accelerometers[7], strain gauges[8], inclinometers[9], total stations[10], fiber Bragg grating sensor[11], laser Doppler systems[12] and laser scanning[13], encounter limitations in field applications owing to factors including prohibitive installation expenses and constraints related to measurement locations. In recent years, there has been remarkable growth in vision-based structural monitoring research[14–16], which can be categorized into intensity-based and phase-based methods. Intensity-based approaches, such as digital image correlation (DIC), optical flow, and advanced computer vision techniques, have been extensively investigated[17–20]. In contrast, phase-based methods like Fourier transform profilometry[21], phase-based motion magnification[22], windowed Fourier transform[23] and sampling moiré (SM) method[24–26], exhibit robustness to intensity noise in images and thus making it as a promising solution for accurate infrastructure inspection. Despite their popularity over the past decade, their reliance on stationary cameras presents a significant limitation for deployment in the field, particularly for critical infrastructures such as bridges that traverse mountainous or aquatic terrain. To surmount the limitations inherent in traditional stationary cameras, unmanned aerial vehicles (UAVs) offer a propitious perspective. Possessing unparalleled maneuverability, UAVs are well-suited to effectively address the bottleneck problems of camera resolution insufficiency and mounting inconvenience. Numerous research endeavors have been dedicated to developing drone-based displacement measurement techniques, yielding noteworthy advances in recent years[27–39]. A comprehensive summary of the current state-of-the-art in drone-based displacement measurement is provided in Supplementary Table 1. Nonetheless, achieving ultra-high accuracy displacement measurements at sub-millimeter level under the hostile conditions of aerial photography remains a formidable challenge for practical inspection applications.

This research presents a framework to address the challenge of achieving ultra-high displacement measurements with comparable results to the most extensively applied deflection measurement sensor by using video footage captured by a drone camera, as shown in Fig. 1. Specifically, we affixed a group of motion trackers, comprising three markers, to the side of the bridge. These trackers record the structural displacement induced by the external load, together with the movements of the drone camera. Notably, this study introduces the SM method to characterize phase information for displacement measurement. The SM method possesses multiple favorable properties, including robustness to intensity noises, high computational efficiency, and an exceptional level of precision as of 1/100 pixels[25,40]. These qualities provide a solid foundation for further displacement characterization. Then, to separate the actual displacement from the camera motion, we developed a bio-inspired technique called the active balancing compensation (ABC) strategy, which is simple yet effective for extracting the target structural displacement. Besides, it renders further advantages including camera calibration-free and ease in parameter setting. We conducted rigorous theoretical analysis and in-field experiments to validate the proposed system. The experimental results reveal that our system attained a level of accuracy that is on par with the conventional deflection measurement sensor. This marks the first report on the feasibility of utilizing a drone photography-based system as an effective solution for practical bridge displacement measurement. We believe that the proposed methodology could play a significant role in shaping the next generation of bridge inspection systems, allowing for autonomous processing, favorable flexibility, and preferred cost efficiency.

## Results and discussion
### Drone-based displacement measurement approach
Drawing inspiration from the balance sensing system of the human inner ear, we proposed a technology that allows for robust and high-precision image de-blurring. Figure 2 depicts the schematic diagram of the proposed approach, which incorporates the utilizing of two 2-D reference markers as receptors within the human ear's vestibular system. This methodology is specifically designed to achieve image stabilization during drone hovering. Here, our focus centers on the medical perspective of the ear's structure, and we draw upon this knowledge to construct our displacement measurement system. As it is well researched, the inner ear comprises receptors for both hearing (the cochlea) and balance (the utricle, saccule, and semicircular canals)[41], as depicted in Fig. 2a. The vestibular receptors are composed of two distinct components: (i) two otolith organs, namely the utricle and saccule, responsible for measuring linear accelerations, and (ii) three semicircular canals, tasked with measuring angular

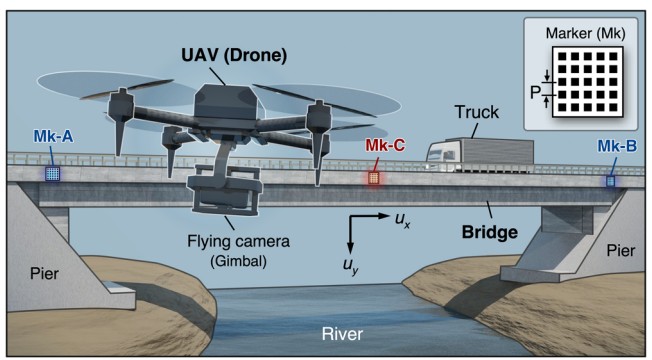

**Fig. 1 | Overview of the measurement system based on drone aerial photography by utilizing the moiré phase analysis methodology.** The developed method simply consists of a commercial drone camera and three markers with a regular grating pitch to measure the deflection of bridges accurately.

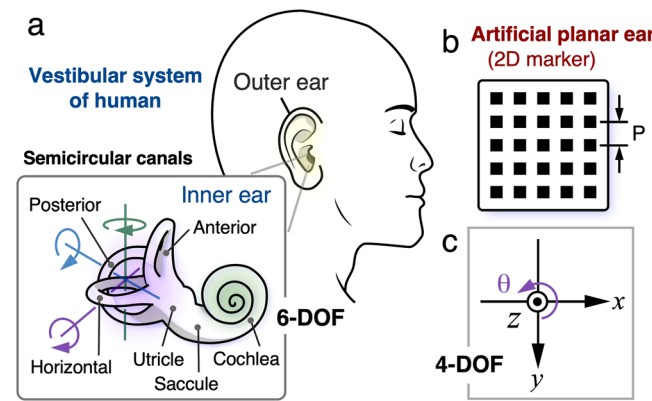

**Fig. 2 | Bio-inspired image stabilization by using two 2-D artificial ears (reference markers) in vestibular system. a** Vestibular system in human being. **b** Artificial planar ear (i.e., 2D marker). **c** The coordinate of 4 DOF, which the translation of $x$, $y$ and $z$ the rotation angle of $\theta$ can be measured from two 2D markers.

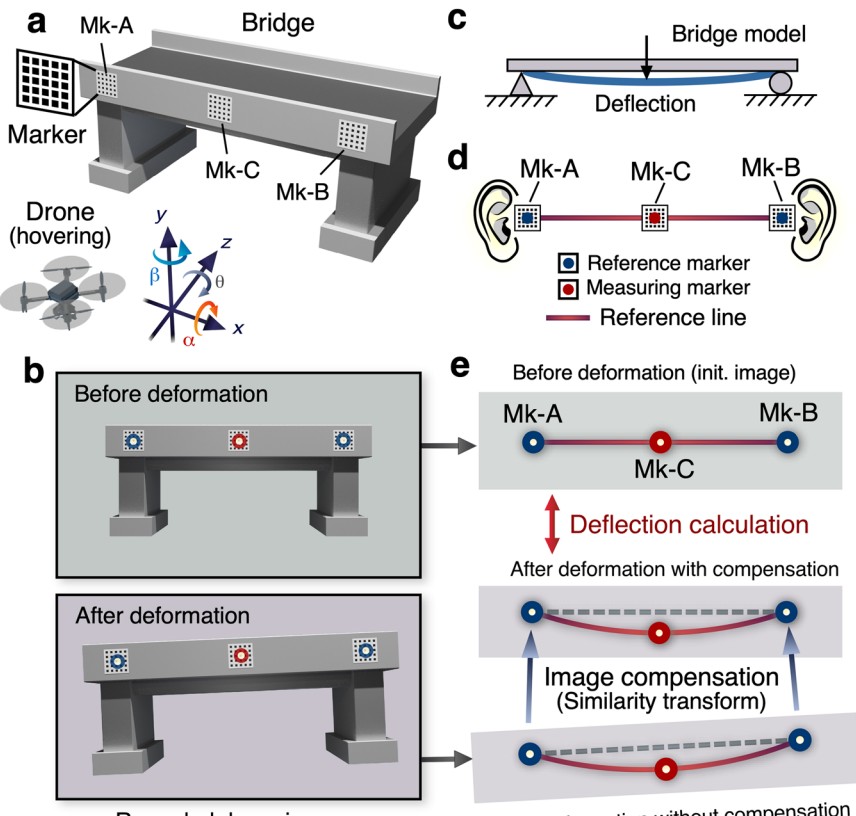

**Fig. 3 | Fundamental principle of the active balancing compensation (ABC) method using a reference line. a** Experimental arrangement of drone deflection measurement. **b** Recorded drone images before and after deformations. Each image contains two reference markers (Mk-A & Mk-B) and one measuring marker (Mk-C). **c** Our assumption according to a general bridge model. There are two fixed points at each end of the bridge (zero displacement or small enough displacement in the *y* direction to be negligible). **d** The key idea of our ABC method using a reference line. The reference line formed by markers Mk-A and Mk-B serves an analogous function to the human balance mechanism of the ear. **e** The post-deformation image compensation involves a dual application of similarity transform, achieving an exceptional 1/100-pixel accuracy.

accelerations. Specifically, rotational motion (angular acceleration) manifests during head rotation, while linear acceleration is perceived during walking, falling (i.e., translations), or head tilts concerning gravity. These receptors transmit vestibular information to the brain, where it is amalgamated into appropriate signals related to the direction and speed of movement, in addition to the head's position relative to gravito-inertial acceleration. Therefore, the ear encompasses five types of receptor and a total of ten receptors in human ear, providing balance on six axes in three dimensions world.

Taking a cue from the ear's sense of balance, in this study, we have developed a simple yet efficient approach to achieve stability in measuring deflection, one of the most critical vital signs in bridge inspection. Specifically, we have characterized the geometric structure of the bridge and hovering camera setting to simplify the six degrees-of-freedom (DoF) into four DoF, which forms the foundation for achieving stability in measuring deflection. To achieve this, we have utilized a two-dimensional pattern of a marker, as depicted in Fig. 2b, can emulate the equilibrium sensing function of the human ear, offering a robust image blur compensation and accurate (in-plane) deflection measurement. In this setup, by placing two reference markers on the abutments at both ends of the bridge, through leveraging the human binaural balancing function, the four DoF displacement measurement error, which includes the translation of *x, y, z*, and the rotation of *θ* (as shown in Fig. 2c), caused by the drone's movement can be automatically canceled.

Figure 3a illustrates the optical configuration for bridge deflection measurement via drone aerial photography, accompanies by the

definition of the six axes of the drone camera. While the drone is in the airborne state of hovering, even minor variations in its position and attitude can cause simultaneous displacements in the *x* and *y* axes ($\Delta x$ and $\Delta y$), magnification along the *z*-axis ($\Delta s$), and rotational adjustments ($\Delta \theta$). In images acquired before and after the bridge's deformation by the inspection vehicle, as depicted in Fig. 3b, two reference markers (Mk-A and Mk-B), analogous to the left and right human ears, form a reference line. The model of the bridge, as adopted in this study (Fig. 3c) assumes insignificantly negligible displacements along the *y*-direction at the fixed points on both ends. For precise image deblurring, markers Mk-A and Mk-B function analogously to the human auditory equilibrium, with a "reference line" connecting the two, as shown in Fig. 3d. A similarity transformation aligns these reference lines in images before and after deformation with an accuracy level of 1/100 camera pixels. The detailed workflow can be referred to in Supplementary Fig. 1. Initially, the efficient maximized cross-correlation (MCC) method[42] is employed to extract the center coordinates of markers Mk-A and Mk-B with pixel accuracy. Subsequently, a similarity transformation[43] is implemented to align the images with pixel precision to the first frame of video before deformation. This step allows us to achieve reliable alignment of the measurement markers before and after deformation within half of the marker pitch, laying the foundation for further ultra-fine sub-pixel analysis using the SM algorithm[24,25]. Subsequently, the phase-based SM method is employed to compute the center coordinates of each marker with enhanced 1/100 pixel accuracy[40] and the accurate marker pitch in image[44]. The critical point of the SM method is mentioned in Supplementary Fig. 3.

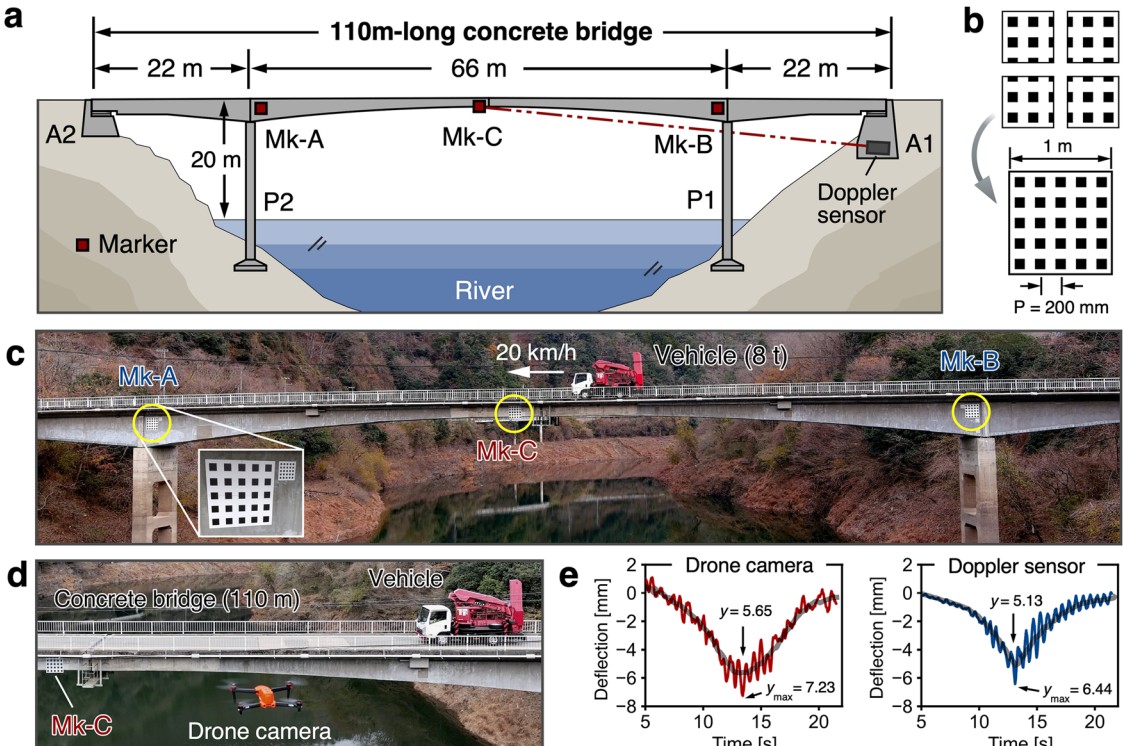

**Fig. 4 | Deflection measurement of a 110-meter-long concrete bridge by using a commercial drone camera. a** Schematic diagram of the target bridge. **b** The specification of the marker with a 200 mm pitch. Four different markers, each 0.5 m square in size, are combined for easy portability. **c** The record image by the drone camera (Autel EVO-II Pro; 6 K movie) from a distance of 85 m. **d** The top-view of a scene during measurement test. **e** The measured results obtained by the developed method (*left*) and conventional Doppler displacement sensor (*right*).

Notably, the SM method has been successfully used in previous studies to measure deflections of bridges using a fixed camera, achieving ultra-high precision on the sub-millimeter accuracy[45–47]. In the present study, we adopt the SM method to extract accurate marker center coordinates, which not only compensates for image blur induced by camera motion but also facilitates highly accurate estimation of deflection. It is imperative to note that the goal of achieving 'perfect' image stabilization is defined as rendering zero displacements between reference markers Mk-A and Mk-B. Consequently, an ABC method using a reference line is proposed to facilitate more precise deflection measurements. This approach is simple yet remarkably potent since it eliminates the displacement errors arising from the residual effects of $\Delta x$, $\Delta y$, $\Delta s$, and $\Delta \theta$ induced by image stabilization procedure automatically. Additionally, the drone's gimbal function significantly reduces out-of-plane rotation ($\Delta \alpha$ and $\Delta \beta$) during hover recording (See Supplementary Figs. 5 and 6). Thus, the proposed methodology enables sub-millimeter accuracy in measuring bridge deflections, even with drone aerial photography, attaining comparable results to using fixed cameras and the SM method in previous research.

### Field experiment of bridge deflection measurement

A field experiment was conducted to assess the effectiveness of the proposed methodology using aerial drone photography on a 110-meter-long concrete bridge, as illustrated in Fig. 4a. This bridge, a Druk-Bund structure (hinged in the center) with three spans, was built in 1959 and located more than 20 meter height from the river. This application showcases the proposed drone-based displacement measurement system, as the bridge is situated above a river, making it impossible to find a location to fix the camera as in conventional vision-based displacement measurement methods. The details of experimental setup, marker installation and attachment, and the details of the

target bridge can be referred to in Supplementary Figs. 7–10, respectively.

The bridge was imaged as a movie using a commercial drone camera while an 8 t test vehicle was traversing the bridge at a speed of 20 km/h. At a distance of 85 m between the aerial drone (Autel Robotics, Autel EVO-II Pro) and the bridge, the 200 mm pitch marker (Fig. 4b) was captured by only 13 pixels in the recorded 6 K (5472 × 3076 pixels) movie (Fig. 4c). The theoretical analysis esti-mated the achieved measurement resolution to be 0.2 mm, which is deemed sufficient to measure the anticipated displacement caused by the 8 t truck's passage across the bridge. The proposed metho-dology therefore achieved a favorable precision in measuring millimeter-scale displacement, and the results were highly con-sistent with the reference results generated by LDV, as depicted in Fig. 4e. The maximum deflection values obtained using our method and the Doppler sensor were measured as 7.23 mm and 6.44 mm, respectively, yielding a discrepancy of 0.79 mm. Similarly, the maximum deflection values after smoothing obtained using our method and the Doppler sensor were 5.65 mm and 5.13 mm, yielding a discrepancy of 0.52 mm. Furthermore, the side-by-side compar-ison results indicate that camera motion issues have been effectively resolved (See Supplementary Fig. 11 for intermediate results of drone-based deflection measurement). To demonstrate the repeat-ability of the proposed approach, we conducted the experiment for multiple times. Furthermore, with the objective of clarifying the limitation of our method, we extended the distance of between the drone and the bridge from 85 m to 100 m. These results can referred to Supplementary Fig. 12. The experimental findings unequivocally demonstrate the aerial photography technique's efficacy in provid-ing an accurate and practical deflection measurement methodology for common bridge structures, thereby offering a viable solution to the long-standing challenge in the field.

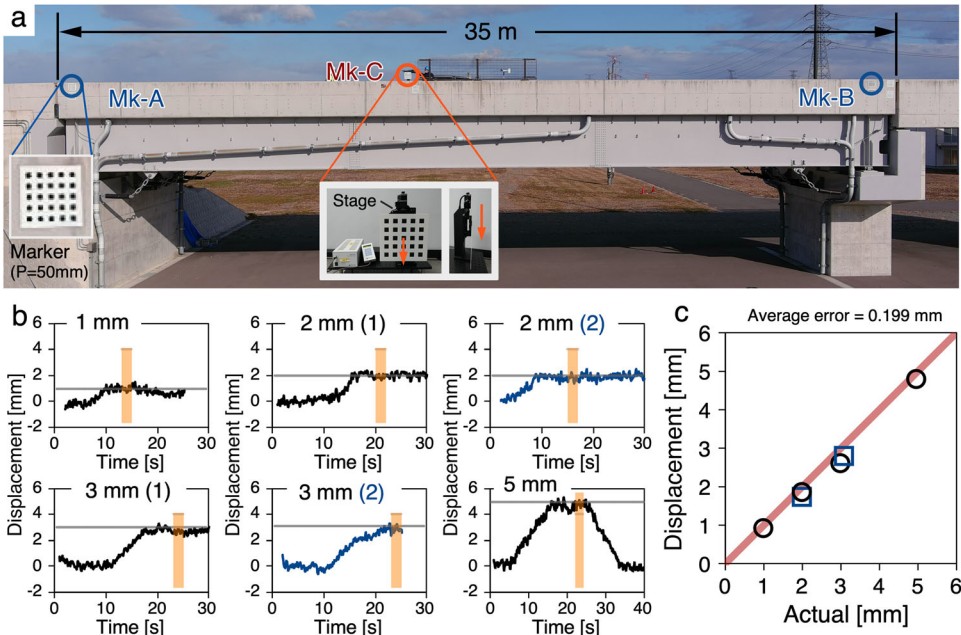

**Fig. 5 | Deflection measurement accuracy through a field experiment for a general 35 m scale concrete bridge. a** Recorded image by drone camera. **b** The measured results. **c** Measurement accuracy.

## Demonstrations of measurement accuracy verification

In light of the fundamental significance of reproducibility and replicability in scientific research, we carried out additional in-field displacement measurement experiments to establish the reliability of our proposed approach. More precisely, we conducted a series of experiments on a 35-m-long bridge situated at the Fukushima robot test field in Japan, wherein we compared the deflection values obtained by our drone photography-based technology with the displacement values in the y-direction, achieved by controlling a marker with a high-precision linear moving stage. The experimental arrangement is depicted in Fig. 5a, and the experimental scene is shown in Supplementary Fig. 13. In this experiment, the grid pitch of all three moiré markers was 50 mm, and the distance between the drone and the bridge was 34 m. The weather (early Jan. 2023) was sunny, the temperature was 5.2 °C, and the wind speed was 0–6.6 m/s. During the experiment, a measuring marker (Mk-C) was placed near the center of the bridge and longitudinally displaced manually by 1.003 mm, 2.010 mm, 2.999 mm, and 4.967 mm using a precision moving stage (SURUGA SEIKI, KX1250C-R; Unidirectional positioning accuracy: <5 μm; resolution: 0.05 μm/pulse). The resulting displacements were captured using drone photography to enable a comparative assessment of the accuracy of the proposed method. To evaluate the reproducibility of our approach, we conducted two measurements at 2.000 mm and 3.082 mm displacements, respectively. The measurement results obtained through the developed technique are indicated in Fig. 5b, where the mean value for a one-second interval within the light-orange region is presented. Figure 5c illustrates the correlation between the measured and actual displacements, with an average error of 0.199 mm, surpassing the accuracy of previous studies on drone-based displacement measurements.

The two validation experiments have also provided evidence of the exceptional flexibility of our approach, as it enabled the precise measurement of sub-millimeter level displacements for target bridge structures of varying sizes and spans by adjusting the marker pitch (from 50 mm to 200 mm) and the distance (from 30 m to 85 m) between the bridge and drone during image capture. Our approach provides a reliable and innovative methodology for practical bridge displacement measurement with the potential to significantly impact the development of future autonomous bridge inspection systems. On the other hand, our proposed method also has its limitations: (i) Since displacement measurement accuracy is tied to the chosen marker pitch, therefore, measuring minute displacements beyond 1/1000th of the marker pitch proves challenging based on the accuracy of the sampling moiré method. (ii) Our proposed method assumes the fixed girders remain stationary or experience negligible displacements, rendering bridges with unique structure unsuitable for this approach. (iii) The measurements are susceptible to several critical weather-related factors such as strong wind and shimmer for long-distance image shooting under unfavorable weather. For an in-depth discussion, please refer to Supplementary Note 2.

In conclusions, we have demonstrated that displacement measurement of infrastructures at a millimeter scale by using drone photography is feasible. Our method encompasses an ordinary off-the-shelf drone, a set of markers, and a well-designed computational framework for high-precision displacement extraction. As a vision-based system, our proposed approach differs from conventional methods, relying on intensity information. Instead, we characterize the phase information to estimate subtle displacements, providing multiple favorable features, including high accuracy, low computational complexity, and robustness to pixel noise and visual artifacts. Furthermore, to eliminate the errors induced by drone camera motion in displacement measurements, we proposed a simple yet efficient four DoF parameterization scheme taking advantage of the geometry constraints of drone photography of bridges. As a result, we successfully relaxed the requirement of recovering 6 DoF and camera intrinsic parameters. This simplification facilitates computational analysis and enhances displacement measurement stability by suppressing parameter estimation errors.

Through rigorous lab-scale experiments and extensive real-field validations, we have showcased that our approach achieves accuracy on par with conventional displacement measurement sensors, while simultaneously providing superior flexibility, reduced cost, and ease of onsite installation. Our proposed methodology holds great potential for widespread use in a variety of fields, including infrastructures, heavy industry plants, and construction sites, and is primed to make a significant impact in the expansive field of drone inspection[48,49]. Last

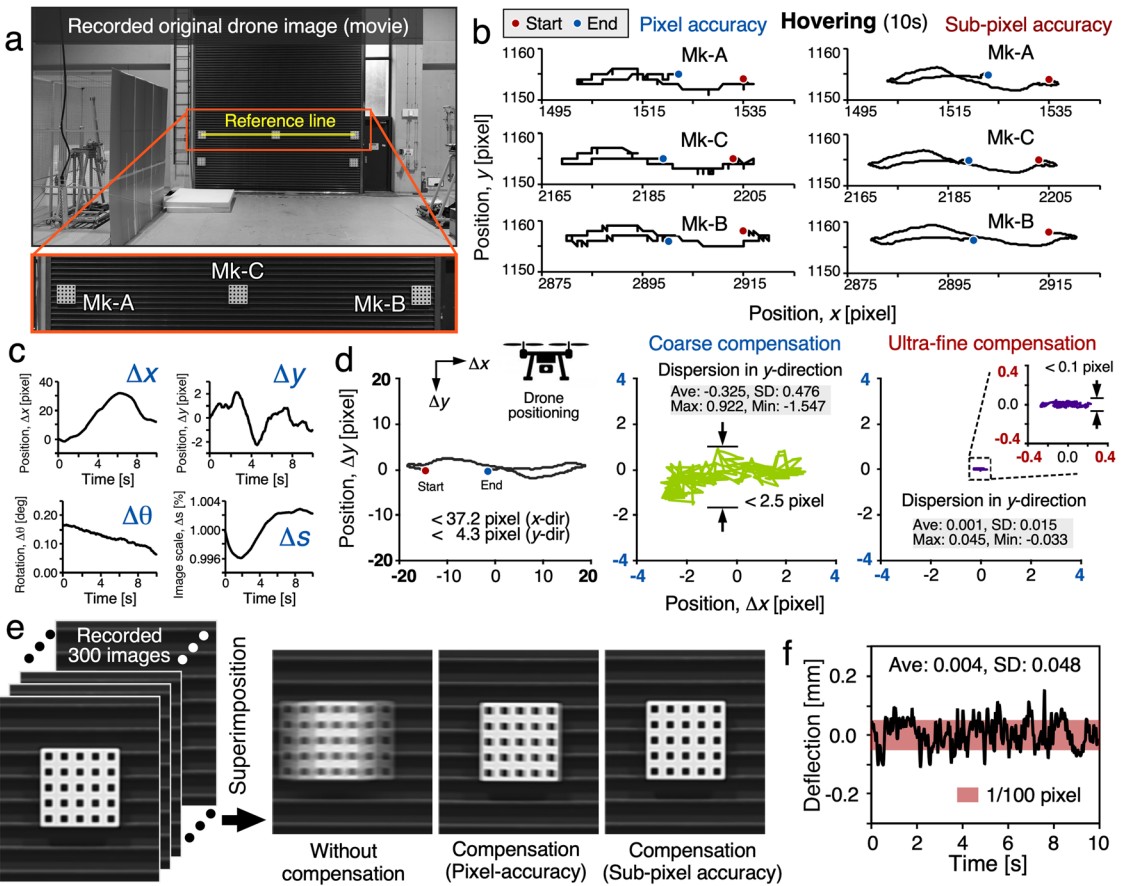

**Fig. 6 | Image de-blurring compensation. a** Recorded image by a drone camera (DJI, Inspire 2). Image blurring compensation is achieved via a reference line established from the center coordinates of Mk-A and Mk-B. **b** Measured trajectory of each marker with pixel accuracy (*left*) and sub-pixel accuracy (*right*). **c** calculated drone movement by using the reference line. **d** Image blurring results for (*left*) without compensation, (*center*) first compensation with pixel accuracy, and (*right*) second compensation with sub-pixel accuracy. **e** Comparison images when 300 captured images are superimposed: without compensation (*left*), compensation with pixel accuracy (*center*) and compensation with sub-pixel accuracy (*right*). **f** Final measured displacement result during 10 s hovering.

but not least, this research holds immense potential for future autonomous inspections, reducing human intervention and improving efficiency. Through this automation, regular and timely inspections van be conducted while maintaining a high level of safety, effectively addressing the challenges posed by aging infrastructures[50].

## Methods
### Marker center tracking
Our analysis commences with the extraction of coordinate trajectories for each marker, encompassing both the drone camera's motion and the load-induced structural bridge displacement. The $x$- and $y$-axis coordinates are obtained using the MCC method, which efficiently and effectively tracks marker movement between adjacent frames in the drone inspection video. Figure 6a shows a representative example in which motion information can be readily extracted by repeatedly searching for this location between the 2D DFT of the template marker image in the initial frame and marker images in the sequence. After motion extraction using the MCC-based algorithm, we obtain the motion trajectory of markers across the video footage at pixel-level accuracy, as shown in left side of Fig. 6b. Additionally, the angles induced during marker installation can affect displacement precision. To mitigate this factor, we developed a dedicated algorithm to rectify the initial angle issue, as elucidated in Supplementary Fig. 2.

### Coarse-compensation with pixel-level precision
A drone camera is commonly equipped with a gimbal to stabilize the camera during flight, allowing for smoother and more stable footage.

Although it prevents the footage captured by a drone camera from being shaky or blurry in aerial photography and videography, the camera motion would inevitably be reflected in the pixel values of captured videos and overwhelms the subtle displacement of structures. To compensate the camera motion trajectory for precise displacement measurement, we propose a simplified approach that reduces the 6 DoF problem to 4 DoF by considering two on-site parameter constraints: the planar structure of the bridge side supported by the markers and the hovering mode used for aerial photography. Specifically, we utilize two reference markers to obtain four parameters that govern the degree of freedom in a given system: motion along the $x$ and $y$-axes ($\Delta x$, $\Delta y$), angle of rotation ($\Delta \theta$), and image scaling ($\Delta s$), as shown in Fig. 6c. The effectiveness of image blurring compensation in the 4 DoF case is confirmed by simulation (See Supplementary Fig. 4). Following the aforementioned analysis, we utilize a pixel-accuracy similarity transform[43] to align the entire drone inspection video with the initial frame, using the center coordinates of all tracked markers. Supplementary Note 1 provides further details. As result, the shifted markers in the video sequence are then determined to be within half the marker pitch distance from the marker template in the initial frame, allowing for phase-based moiré analysis to retrieve the sub-pixel coordinates of marker centers, as depicted on the right side of Fig. 6b.

### Displacement measurement by the sampling moiré method
We utilized the SM technique as a fast, simple, and accurate method for measuring small-displacement distributions (See Supplementary

Fig. 3). The approach involves analyzing digital images of a regularly repeating pattern with a known pitch on the structure, which allows for determining in-plane displacements across the structure. Upon implementing the image processing of down-sampling and intensity interpolation, multiple phase-shifted moiré fringes can be generated simultaneously. Subsequently, the phase distribution of the moiré fringe can be determined using the phase-shifting method. Finally, measuring the phase differences between the moiré fringes before and after deformation, can accurately determine the displacement distribution with a precision of 1/100 pixel or 1/1000 of the grating pitch. In contrast to the DIC method, which uses a random pattern, the SM technique leverages a regular high-contrast grating pattern on the structure, thereby decreasing the computational complexity of the image processing.

**Ultra-fine compensation at one-hundredth pixel level precision**
The ultra-high precision 1/100 pixel marker center coordinates extracted by the SM analysis are utilized to calculate the subtle displacements induced by camera motion. This step utilizes a similarity transformation, similar to coarse-compensation but integrating moiré analysis to acquire marker center coordinates with a precision of 1/100th of a pixel. This notably enhances the accuracy of estimating the 4 DoF similarity transform parameters and significantly improves aerial image deblurring performance. The effectiveness of this approach is illustrated in Fig. 6d, e and demonstrated in Supplementary Movies 1 and 2. As a result, we attain an the *y*-directional displacement can be measured with an accuracy of 1/100 of a pixel (Fig. 6f).

## Data availability
All data supporting the findings of this study are available within the article and its supplementary files. Source data are provided with this paper.

## Code availability
The source code for this study is registered and managed by the National Institute of Advanced Industrial Science and Technology (AIST). It could be made available for disclosure upon acceptance of AIST's terms and conditions. For code-related inquiries, please contact Shien Ri and Jiaxing Ye.

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

## Acknowledgements

The authors thank Dr. Tetsuya Yamamoto and Dr. Peng Xia for assistance with the field experiment, and Ms. Yuri Noguchi for the figure preparation. We also extend our appreciation to the members of the Laboratory on Innovative Techniques for Infrastructures (ITIL) at Kyoto University for their collaboration in conducting this study.

## Author contributions

S.R. and J.Y. contributed to the study idea and performed the data visualization and wrote the paper; S.R. and N.O. conceived and designed experiments; N.O., S.R. and J.Y. performed the experiments and data acquisition; J.Y. and S.R. implemented the computer code and supporting algorithms; conducted data analysis and results validation; N.T. provided leadership of this research project. All the authors read and contributed to the discussion and provided feedback on the manuscript.

## Competing interests

The authors S.R., J.Y. and N.T. declare the following competing interests: the authors S.R., J.Y. and N.T. are filing a Japanese patent application entitled displacement measurement method, image correction method and program thereof (P2023-82603A) for the algorithm described in this paper. The remaining author declares no competing interests.
