## [Peer Review File · Nature Communications]

Drone-based displacement measurement of infrastructures utilizing phase informationREVIEWER COMMENTS

Reviewer #1 (Remarks to the Author):

The paper proposes a framework that uses drone cameras to achieve highly accurate displacement measurements. The idea -and actual deployment- of drones for this purpose is not new (as acknowledged by the table of such deployments) so the added value/novelty is naturally in the specific methodology.

The challenge (of drone-based displacement measurement) lies in the algorithm for compensating for errors induced by drone camera movement during hovering.

This is where the novelty lies. The sampling moiré technique, previously developed by the authors, in the proposed framework has been published in other journals (this paper has cited the relevant literatures).

The application is indeed very impressive and the resolution good. This is the first time I've seen drone-based displacement measurement used for concrete bridges with such a large field of view; the camera covers a span of a bridge that extends over 60 meters. I believe that the proposed measurement framework is revolutionary, and its effectiveness has been validated in a field test.

I don't think the proposed drone-based method has significant superiority over the ground camera-based method using the sampling moiré technique. This is because the proposed method requires the installation of (rather large) artificial targets and a large field of view from the camera (which is also a benefit) to cover the two reference markers, which are placed separately at the two ends of the girder. These requirements diminish the advantages of drone-based measurement. But if you have to use drones (because you can't locate on ground) but you can place markers/targets this is a demonstrably powerful methodology.

Some specifics:

Because this is a Nature journal I'm being more pedantic than I would for more specialist journals. First three sentences of the abstract are a very clunky introduction with much redundancy. I suggest blending a couple of phrases from it into the 4th sentence, while 5th sentence has redundant 'the's. Later "utilize efficacious and cost- effective" is a starkly contrasting style that might fit in a verbose powerpoint. The abstract should do a better job.

Issue of aging infrastructure ... recent years. Well, yes, the more recent, the more the more aged the structure!

It doesn't automatically mean the bridge is deficient, however natural or manmade actions may lead to 'damage' or degradation. It would be better to focus on e.g. ASCE infrastructure report card or similar EU/Asian exercises that specifically look at condition, then the need becomes more stark. The point about consequent economic loss is lack of resilience which is not just that bridges fail but that they are taken out of service for a length of time, and efficient and effective inspection can inform structural intervention solutions to minimise total loss. It's not just about spotting 'damage' by inspection but looking for performance anomalies that signify deep problems you can't see (think of human internal or external bleeding). So your point about furnishing indispensable insights is spot on.

"Recent studies have focused on vision-based structural health monitoring methods for assessing civil infrastructure". True, but better to say that is it is a relatively hot and fast growing research area in structural health monitoring (SHM).

Reliance on stationary cameras is not quite right as it's well recognised that there is no such thing in the real world as a fixed camera; even total stations use fixed references to compensate for instrument movement and some of your references should (if not already) point to methods for removing camera shake effects from vision system data. The challenge is to translate the compensation strategies to a bigger scale.

The explanation of the compensation methodology is not entirely clear. The compensation process, which is critical for readers, should be described using formulae. I'm uncertain whether the authors incorporate the parameters (4DOF) of the two reference markers for image compensation, or if they are compensating each reference marker and measuring target separately.

The proposed compensation methodology requires two reference markers to be attached separately at two stationary areas, and the measuring targets should be within the range of these two references. In the field test, these two reference markers were installed at the ends of the girder, close to the piers. This could pose a significant limitation to its application.

Exactly how hostile are real-world conditions of aerial photography? So on page 3 can you define the 'standard deflection measurement system/conventional deflection measurement sensor' and how reliable that is as the 'gold standard'? 'Credible alternative' is indeed the yardstick, but how to define this? -this is rather a vague statement.

In Fig 2b will your drone always be looking directly at the pattern i.e. not oblique with missing DOFs assumed irrelevant? The further away the less oblique (i.e. more perpendicular) the angle but the poorer the resolution; what is the tradeoff here? You don't show ' $\Delta\alpha/\beta$ ' in figure 3, or 'LINE' (why caps? As opposed to 'as a line') and how are mk-A/B shown as 'A' and 'B' on the figure? You're not describing the figure too well. 'Normalised Cross-Correlation' is not a proper noun and so only the acronym should use capitals, look for similar misuse elsewhere e.g. with 'ABC'.

So a 'marker' is an assembly of black squares on a grid; this is often referred to as a 'target'; the black squares could individually be targets or 'salient features'. There is no standard terminology but perhaps a look at what's conventionally used vs what you used would be helpful to you and readers. It seems the periodic pattern is essential; are there other target/marker/feature patterns that could work or is this fundamental, so limiting to specific artificial markers/targets? This, and the size of the marker/target warrant some discussion.

'20m from the river' means above the river? For consistency, as you use 'mm' to quantify pitch and 'km' for speed, why not use 'm' throughout to quantify length and distance? Also 'ton' is not an SI unit, use metric ton (tonne or 't'). Figure 4 use non-breaking spaces where appropriate in the captions, although type-setting this may be fixed.

So with the 35 m bridge, it's not about bridge displacement but a precision-controlled movement relative to a deck that was (reasonably) assumed not to move?

'As a vision-based system, our proposed approach differs from conventional methods, relying on intensity information' -does this mean methods that rely on ?

So is this a revolutionary approach? I think there is enough to say 'yes'. One drawback is the need to artificial targets (or markers), whereas natural targets allow tracking at inaccessible locations -a common feature of a lot of infrastructure that would be ideal for this type of technology -see general comments above/

Supp info. You mean tilt not tile angle? You use it twice and maybe the collection of squares is a 'tile' -indeed it looks like one, but you don't use the term in the main document.

Reviewer #2 (Remarks to the Author):

The paper presents a pipeline for drone-based displacement measurements using phase-based sub-pixel corrections. The paper is well written, and the supplementary results are impressive. However, the paper makes key assumptions which should be clearly articulated. The paper also fails to reference several highly-cited and related topics and as a result makes inaccurate claims.

The following pieces of research on using drones for displacement measurement have not been cited. This is just a sample list, there are other papers too that have been missed.

- Bolognini, Michele, et al. "Vision-based modal analysis of built environment structures with multiple drones." *Automation in Construction* 143 (2022): 104550.
- Hoskere, Vedhus, et al. "Vision-based modal survey of civil infrastructure using unmanned aerial vehicles." *Journal of Structural Engineering* 145.7 (2019): 04019062.

- Weng, Yufeng, et al. "Homography-based structural displacement measurement for large structures using unmanned aerial vehicles." *Computer-Aided Civil and Infrastructure Engineering* 36.9 (2021): 1114-1128.

The statements in the manuscript should be appropriately modified to reflect the results from recently published research. For example, this statement may have to be reworded. "Although some recent investigations have ventured into outdoor verification experiments, their accuracy has been constrained, with reported root mean square errors (RMSE) of approximately 2 mm at shooting distances of roughly 5 m"

There has also been a large amount of research on phase-based displacement measurement and modal analysis that has not been cited. E.g.,

- Yang, Yongchao, et al. "Blind identification of full-field vibration modes from video measurements with phase-based video motion magnification." *Mechanical Systems and Signal Processing* 85 (2017): 567-590.

The proposed research should be put into the appropriate context after a more detailed literature review.

Details about how the initial angle correction are applied should be provided – is this applied to each reference marker separately? If so, what algorithm is used to automatically detect the marker? Additionally, how are different rotations of the markers reconciled? Is this only applied on the first frame?

The algorithm seems to have an assumption that two stationary points must be available for applicability. This should be emphasized in the abstract and introduction. If this is not the case this should be clarified. Will the algorithm work if the reference markers are out of plan with the measurement marker?

How is the proposed method affected by out-of-plane rotation of the drone (i) at the start of the data capture, and (ii) during the data capture? Similarly, how does the in-plane rotation of the gimbal affect the algorithm? Some clarification on this would be helpful.

Reviewer #3 (Remarks to the Author):

In the manuscript, the authors proposed a drone-based displacement measurement method. The obtained results should be significant and have a good potential for promoting the application of the traditional sample moiré methods. However, before the manuscript can be accepted for publication, the authors should address the following item:

1. In the sampling moiré method, the obtained phase information is relative value which requires manually specifying the zero point of the phase. However, marker C moves with the bridge as a whole and there is no zero point of displacement(phase) in this area. How to choose the zero point of the phase?

2. In the part of "Methods, Coarse-compensation with pixel level precision", similarity transform is implemented to shift the markers in the video sequence to be within half the marker pitch distance from the marker template in the initial frame. The step is crucial since the subsequent sampling moiré method utilizes relative phase which means any displacement greater than half the marker pitch distance may cause multiplicity. However, how can the authors guarantee the markers will be shifted within the distance in half marker pitch (from the marker template in the initial frame)?

3. How can the authors fabricate the marker on the bridge structure? Is there any influence from the deviation of the specimen marker from the desired position on the measurement results?

Response to Reviewer Comments

Dear Reviewers

Thank you for your thorough review and valuable feedback on our manuscript. We greatly appreciate the supportive comments, such as “*The proposed measurement framework is revolutionary* (Reviewer #1)”, “*The paper is well written, and the supplementary results are impressive* (Reviewer #2)”, “*The obtained results should be significant and have a good potential* (Reviewer #3)”, and the reviewers’ constructive comments for further enhancements regarding the importance of infrastructures inspection and prospects, the validity of the 4 DoF model, and details of phase analysis techniques. Consequently, we have carefully considered all comments and made the necessary revisions to address them comprehensively. Below, you’ll find our point-by-point responses to the comments, with the Referees’ remarks presented verbatim and our responses in **blue texts**.

Reviewer #1 (Remarks to the Authors)

General comments: The paper proposes a framework that uses drone cameras to achieve highly accurate displacement measurements. The idea -and actual deployment- of drones for this purpose is not new (as acknowledged by the table of such deployments) so the added value/novelty is naturally in the specific methodology. The challenge (of drone-based displacement measurement) lies in the algorithm for compensating for errors induced by drone camera movement during hovering. This is where the novelty lies. The sampling moiré technique, previously developed by the authors, in the proposed framework has been published in other journals (this paper has cited the relevant literatures).

The application is indeed very impressive and the resolution good. This is the first time I’ve seen drone-based displacement measurement used for concrete bridges with such a large field of view; the camera covers a span of a bridge that extends over 60 meters. I believe that the proposed measurement framework is revolutionary, and its effectiveness has been validated in a field test.

Reply for general comments: We are grateful for the constructive suggestions and insightful comments. During the revision process, we have carefully addressed all comments to further improve the quality of our manuscript. As acknowledged by the reviewer, our major contribution to this research is developing an efficient approach to delineating camera motions to achieve sub-millimeter

precision displacement measurement. Fundamentally, it involves two supporting techniques: First, from the viewpoint of optical imaging measurement, we adopted the sampling moiré phase analysis technique, which enabled sub-millimeter precision measurement with 1/1000 pitch (or 1/100 pixel) accuracy. High-precision input guarantees all further processes, including both camera motion estimation and genuine bridge displacement measurement. Second, to estimate camera motions, we employed the 4 degree-of-freedom (DoF) formulation, which had been proved to be simple yet sufficient for this application. Theoretically, drone motion is subjected to 6 DoF non-stationary motion and therefore the model has been extensively investigated in previous studies. However, we argue that a 4-degree-of-freedom (4 DoF) approximation is applicable for the case of drone hovering. Our approach and design have been supported by both theoretical analysis, which includes computer simulation (see **Supplementary Figure 4**), as well as validation experiments conducted in the laboratory (see **Supplementary Figures 5 and 6**) and in-field experiments (see **Supplementary Figure 12**). Combining the two ideas together, we achieved sub-millimeter precise displacement measurement using drone photography. Following the successful validation of this technique through rigorous verification experiments using real bridges, for the first time, this paper showcases that vision-based UAV inspection can achieve comparable accuracy in measuring bridge displacement as the commonly used Doppler sensor method.

Additional general comment: I don't think the proposed drone-based method has significant superiority over the ground camera-based method using the sampling moiré technique. This is because the proposed method requires the installation of (rather large) artificial targets and a large field of view from the camera (which is also a benefit) to cover the two reference markers, which are placed separately at the two ends of the girder. These requirements diminish the advantages of drone-based measurement. But if you have to use drones (because you can't locate on ground) but you can place markers/targets this is a demonstrably powerful methodology.

Reply for additional general comment: We thank the reviewer for the comments regarding the drone-based method and ground camera-based method. These comments align with the specific application scenarios that the technique addresses. We would like to clarify that our intention is not to *replace* the fixed camera-based methods with the proposed drone-based measurement technique. Instead, it serves as a complementary tool to compensate for situations where finding a static position to fix the camera is not feasible, such as bridges spanning over rivers and ravines. These scenarios present challenges in terms of camera placement, and our technique aims to fill in those gaps.

Moreover, compared to the fixed camera-based method, drone-based bridge inspection could be advantageous in flexibility and cost efficiency, enabling future autonomous inspection.

As highlighted in the comments, the laborious aspect of the proposed approach lies in setting the markers. Nevertheless, considering that other displacement sensor methods necessitate the attachment of reflective sheets and installation of accelerometers, the labor demands on field workers remain similar to conventional methods. Fundamentally, this arrangement is premised on a geometric abstraction of the typical structural design of bridges, encompassing the pier, girder, and deck. To measure the displacement of the entire bridge structure, it is necessary to place two reference markers at both ends of the girder, with a testing marker set near the center. Future considerations will involve efforts to simplify this process for practical applications.

Comment (1): First three sentences of the abstract are a very clunky introduction with much redundancy. I suggest blending a couple of phrases from it into the 4th sentence, while 5th sentence has redundant 'the's. Later "utilize efficacious and cost- effective" is a starkly contrasting style that might fit in a verbose powerpoint. The abstract should do a better job.

Reply for comment (1): It was very helpful and informative. Based on the reviewer's advice, we revised the Abstract significantly to make it more concise without redundancy and to show the importance of social infrastructure and the significance of this research. We also paid attention to other "the" and wording and made the Abstract as readable to the reader as possible.

[Response] We have revised the abstract following the reviewer's suggestion. The current version is more concise and clearer in conveying the intended information.

Comment (2): Issue of aging infrastructure ... recent years. Well, yes, the more recent, the more the more aged the structure! It doesn't automatically mean the bridge is deficient, however natural or manmade actions may lead to 'damage' or degradation. It would be better to focus on e.g. ASCE infrastructure report card or similar EU/Asian exercises that specifically look at condition, then the need becomes more stark.

The point about consequent economic loss is lack of resilience which is not just that bridges fail but that they are taken out of service for a length of time, and efficient and effective inspection can inform structural intervention solutions to minimise total loss. It's not just about spotting 'damage' by inspection but looking for performance anomalies that signify deep problems you can't see (think of human internal or external bleeding). So your point about furnishing indispensable insights is spot on. "Recent studies have focused on vision-based structural health monitoring methods for assessing

civil infrastructure”. True, but better to say that it is a relatively hot and fast growing research area in structural health monitoring (SHM).

Reply for comment (2): Thank you for raising the important point regarding a more precise description of the current status of the aging infrastructure. We strongly agree with this and have incorporated your suggestions in the manuscript. In detail, we have done the following two parts:

Part-1. In the introduction section, we elucidated that the challenge of aging infrastructure globally is two-fold: bridges nearing the end of their service period and bridges experiencing condition degradation due to heavy usage or environmental factors. Drawing on information from the ASCE, as well as EU/Asian report cards, we provided a comparative review of these two categories in the introduction section.

Part-2. We present an in-depth/comprehensive discussion on the multifaceted negative impacts of aging infrastructure, which include safety risk for inhabitants, time-cost for repairment/replacement, economic losses due to out-of-service period and broader social concerns. There is a crucial need for high-precision and efficient inspection techniques to detect subtle signs of deterioration, thus enabling early remedial action to effectively manage the increasing number of aging structures.

[Response] In response to the reviewer's comments, we have elaborated on the survey portion in the **Introduction**, highlighting the societal challenges brought about by aging and deficient infrastructures. We have also underscored the critical need for efficient and effective inspection techniques to address the pressing issue of worldwide infrastructure assessment.

Comment (3): Reliance on stationary cameras is not quite right as it's well recognised that there is no such thing in the real world as a fixed camera; even total stations use fixed references to compensate for instrument movement and some of your references should (if not already) point to methods for removing camera shake effects from vision system data. The challenge is to translate the compensation strategies to a bigger scale.

Reply for comment (3): As the reviewer points out, even a camera fixed on a tripod is affected by ground vibrations in outdoor field experiments, resulting in slight image blurring. Therefore, almost all sensor-based or image-based measurements use a fixed reference point to calculate the relative displacement to a reference point, thereby reducing measurement errors due to camera vibration. In

fact, in our previous study [Ref. 46], when we experimented with the sampling moiré method using a fixed camera to measure the deflection of a concrete viaduct on a Japanese bullet train (i.e., SHINKANSEN) at a speed of 320 km/h, a single reference marker was placed on the bridge girder and the relative displacement between the measurement marker and the reference marker in the center of the bridge.

In this research, instead of finding “reference *point*” we devise to establish “reference *line*” by using two reference markers placed on the edges of bridge. The displacement can be subsequently measured by calculating the deviation between the horizontal coordinates of a measurement marker attached near the bridge center and the “reference line”. Taking advantage of the sampling moiré method, efficient motion compensation with a similarity transformation can be achieved that aligns these reference lines in images before and after deformation with an accuracy level of 1/100 camera pixels. This is the fundamental model of the developed UAV displacement measurement system, and the details can be referred in **Fig. 3**.

Comment (4): The explanation of the compensation methodology is not entirely clear. The compensation process, which is critical for readers, should be described using formulae. I’m uncertain whether the authors incorporate the parameters (4DOF) of the two reference markers for image compensation, or if they are compensating each reference marker and measuring target separately.

Reply for comment (4): We sincerely appreciate the reviewer for providing valuable feedback on this matter. We apologize for the insufficient explanation of the motion blur compensation method, which is indeed a critical component of our proposed system. Taking this chance of manuscript revision, we have added detailed explanations to clarify the why and how of the motion compensation method based on 4DoF modeling in this application. First, we present theoretical and experimental analysis results of 4DoF model in **Supplementary Fig. 4, Fig. 5 and Fig. 6**, proving that the 4 DoF is sufficient for displacement investigation using drone hovering video. Subsequently, we present the detail process, that is, similarity transformation is employed to align the entire drone inspection video to the initial frame, utilizing the center coordinates extracted from multiple tracked markers. Through this process, we ensure that the acquired marker images align within half the marker pitch distance from the reference frame. This alignment is vital for conducting phase-based moiré analysis and accurately recovering the sub-pixel coordinates of the marker centers. It serves as a fundamental step to counteract displacements caused by camera motion, enabling precise measurement of structural displacements. The detailed formulas and calculations are provided in **Supplementary Note 1** for further reference.

[Response] Theoretical and experimental analysis of 4DoF modeling is presented in **Supplementary Fig. 4** and **Fig. 5**, respectively. We added explanations to the image blur compensation method in “**Methods - Coarse-compensation with pixel-level precision**” section and the detail 4 DoF modeling approach including math formulas are presented in **Supplementary Note 1**.

Comment (5): The proposed compensation methodology requires two reference markers to be attached separately at two stationary areas, and the measuring targets should be within the range of these two references. In the field test, these two reference markers were installed at the ends of the girder, close to the piers. This could pose a significant limitation to its application.

Reply for comment (5): We appreciate your feedback and agree with your point, and placing the two reference markers requires a broader field of view for image recording. Nevertheless, to accomplish this extremely challenging task of detecting sub-millimeter deflection values from aerial photography, the answer we arrived at this moment was that two reference markers, which serve as a sense of balance equivalent to two human ears, are indispensable. Thanks to this idea, as shown in the results of this study's applied experiments, we obtained results in good agreement with conventional sensors for deflections of a few millimeters, even when shooting from 85 and 100 meters. These field test results validated the proposed system is viable for use on major bridges.

Comment (6): Exactly how hostile are real-world conditions of aerial photography? So on page 3 can you define the ‘standard deflection measurement system/conventional deflection measurement sensor’ and how reliable that is as the ‘gold standard’?

‘Credible alternative’ is indeed the yardstick, but how to define this? -this is rather a vague statement.

Reply for comment (6): Thank you for pointing out the practical issue regarding the usage restrictions of the proposed drone-based displacement measurement method. As a measurement system based on optical sensors, our system is vulnerable to factors like fog and strong winds that can increase measurement variability. Therefore, it is desirable to conduct experiments in clear, low-wind conditions to minimize these effects. For the other requirements, such as adverse weather conditions, and safety concerns, we regard our technique adheres to the existing regulations governing common drone-based inspection methods. Sorry for the ambiguous expression of ‘standard deflection measurement system/conventional deflection measurement sensor’ on page 3. We had rephrased the

words to “most extensively applied deflection measurement technique”, eliminating the unclear presentation.

We found that the expression 'credible alternative' in the original manuscript was overstated. Therefore, the term 'credible alternative' in the manuscript was revised to 'an effective method' for accuracy.

[**Response**] We have made revised the expression of ‘standard deflection measurement system” on page 3 to “the **most extensively applied** deflection measurement technique”, ensuring clarity and accuracy. Also, we changed 'credible alternative' to '**an effective method**', to accurately describe the proposed approach.

Comment (7): In Fig 2b will your drone always be looking directly at the pattern i.e. not oblique with missing DOFs assumed irrelevant? The further away the less oblique (i.e. more perpendicular) the angle but the poorer the resolution; what is the tradeoff here?

You don't show ‘delta-alpha/beta’ in figure 3, or ‘LINE’ (why caps? As opposed to ‘as a line’) and how are mk-A/B shown as ‘A’ and ‘B’ on the figure? You're not describing the figure too well. 'Normalised Cross-Correlation' is not a proper noun and so only the acronym should use capitals, look for similar misuse elsewhere e.g. with ‘ABC’.

Reply for comment (7): We thank the reviewer for point out this critical issue regarding drone-bridge relative position and also for the following constructive comments. The question directly related to the fundamental design of this displacement measurement system. We are sorry that we didn't present the content clearly and we take this chance to making further explanations. In the basic setting of our drone-based system, the drone hovers while facing the testing marker (marker-C, centrally positioned on a bridge). In practice, obliqueness can arise due to wind, slight movement during hovering and gimbal functionalities. Thankfully, the sampling moiré (SM) is not confined to capturing images exclusively from a frontal perspective of the bridge. Previous research has validated the method's capability to deliver accurate deflection measurements even when capturing images obliquely (e.g., from beneath a bridge or at a diagonal across it). As such, there is no substantial concern that a slight oblique angle will significantly impact measurement accuracy.

To demonstrate our method's robustness against these oblique angles, theoretical analysis and experimental results have been added to **Supplementary Figs. 4, 5, and 6**. As a result, we clarified that the oblique angle, represented by a composition of alpha/beta angles, will not introduce significant errors to our displacement measurements. In addition, as the drone moves further away

from the bridge, the image resolution of markers decreases, which can lead to poor measurement performance. We have previously addressed this marker image resolution issue in our prior research [pixel limit], revealing that if the marker pitch image exceeds 10 pixels, our phase-based measurement can operate with no difficulty.

We want to express our gratitude for the feedback on Fig. 3 and the inquiry about the "LINE". In light of these comments, we have diligently revised Fig. 3, enhancing its clarity and ensuring its improved comprehensibility. Regarding the "LINE," it is indeed a fundamental aspect of our measurement system. Specifically, the two reference markers fixed on either side of the bridges establish a reference line. This reference line enables the calculation of the relative displacement between the line and the testing marker, which is fixed near the center of the bridge. The capitalization of LINE signifies its significance as the foundational model of our measurement system. Following the suggestions provided in the comments, we have made revisions to ensure consistency in the use of the terms 'A' and 'Mk-A' throughout the manuscript. Also, we have corrected the expression of 'maximized cross-correlation (MCC)' method in **Supplementary Fig. 2**.

[Response] We added **Supplementary Figs. 4, 5, and 6** to investigate the effect on the oblique angle in detail.

Comment (8): So a 'marker' is an assembly of black squares on a grid; this is often referred to as a 'target'; the black squares could individually be targets or 'salient features'. There is no standard terminology but perhaps a look at what's conventionally used vs what you used would be helpful to you and readers. It seems the periodic pattern is essential; are there other target/marker/feature patterns that could work or is this fundamental, so limiting to specific artificial markers/targets? This, and the size of the marker/target warrant some discussion.

Reply for comment (8): We thank the reviewer for the constructive comments regarding the "marker" used in this research. As pointed out by reviewer, it is of great necessity to manifest the terminology we used here which is different from the ones used in previous studies. To clarify the difference, we added **Supp. Fig. 8a** showing a photograph of markers designed and fabricated in experiment. It is noteworthy that the pattern can be either square or circular, as long as the pitch remains regular. The proposed approach can work with artificial markers as if the periodic pattern existed on the markers. We used squares design due to the simplicity in fabrication. In addition, based on the research conclusion that the sampling moiré method's capability to detect minute displacements with a precision of 1/1000 of the grid pitch, the marker size (grid pitch) can be determined with respect to

the desired precision in displacement values of the target bridges.

[**Response**] We added details of marker used in our system in **Supplementary Fig. 8**. In the explanation, we also discussed the relationship between marker design and displacement measurement precision.

Comment (9): '20m from the river' means above the river? For consistency, as you use 'mm' to quantify pitch and 'km' for speed, why not use 'm' throughout to quantify length and distance? Also 'ton' is not an SI unit, use metric ton (tonne or 't'). Figure 4 use non-breaking spaces where appropriate in the captions, although type-setting this may be fixed.

So with the 35 m bridge, it's not about bridge displacement but a precision-controlled movement relative to a deck that was (reasonably) assumed not to move?

'As a vision-based system, our proposed approach differs from conventional methods, relying on intensity information' -does this mean methods that rely on ?

Reply for comment (9): We thank the reviewer for their careful reviewing of the manuscript and their constructive remarks concerning the consistency of units used in the manuscript. The "20m" mentioned in the text represents the distance between the river and the bridge. To provide clarity, we have now included this measurement in Figure 4. We agree that a standardized metric in units is preferred, such as using "meter" for consistency. However, considering the significant disparity in scale between the bridge and the displacements, we have retained "m" for larger structures and "mm" for the precise displacements. Thank you for pointing out the usage of the SI unit 't' and the caption typesetting in Figure 4. We have made the necessary revisions to rectify these issues. In the accuracy verification experiment for the 35-meter bridge, as you correctly described, we attached the marker to a precise linear moving stage to control the displacements accurately in the y-direction, and details can be seen in **Supplementary Fig. 13**. This experimental setup was designed to validate the precision of our proposed approach by simulating various bridge deflections. Since the experiment did not involve any external loads or vehicles, we assumed that the bridge remained stationary, with the displacement at its center equating to zero. We have also rephrased certain statements to enhance clarity throughout the manuscript.

[**Response**] We have incorporated the reviewer's comments and made the necessary revisions to the manuscript. To facilitate easy identification of the updated portions, we have highlighted them in the revised version.

Comment (10): So is this a revolutionary approach? I think there is enough to say ‘yes’. One drawback is the need to artificial targets (or markers), whereas natural targets allow tracking at inaccessible locations -a common feature of a lot of infrastructure that would be ideal for this type of technology -see general comments above.

Reply for comment (10): We would like to express our sincere appreciation for the reviewer's recognition of our research as an innovative technological advancement. In contrast to marker-free methodologies, the technique in question does involve the use of artificial markers, which may be viewed as a drawback. However, this apparent limitation is offset by the inherent benefit of the approach: its reliance on cost-effective markers with a repeating pattern. This characteristic ensures a higher level of reliability in marker tracking and displacement determination, thereby ensuring measurement accuracy.

Furthermore, it holds the potential to emerge as a profoundly effective inspection method for truss bridges and railroad bridges. The feasibility of this concept rests on the availability of naturally occurring regular patterns—such as rivet arrangements in railroad bridges or triangular truss structures in truss bridges. By harnessing these natural (artificial) patterns and foregoing the necessity of marker attachment, the technique could only rely on aerial drone photography for bridge deflection measurement. In anticipation of such scenarios, we have already introduced a methodology capable of measuring displacement using arbitrary repetitive patterns. This approach can be employed seamlessly when an existing natural repetitive pattern on the structure's surface is being evaluated.

Comment (11): Supp info. You mean tilt not tile angle? You use it twice and maybe the collection of squares is a ‘tile’ -indeed it looks like one, but you don’t use the term in the main document.

Reply for comment (11): The typographical error in using "tile" angle instead of the correct term "tilt" angle has been rectified in the revised manuscript. We greatly appreciate your attentive review.

[Response] The mistypo “tile” has been corrected to “tilt” in Supplementary Information (Supplementary Fig. 2).

Reviewer #2 (Remarks to the Authors)

General comments: The paper presents a pipeline for drone-based displacement measurements using phase-based sub-pixel corrections. The paper is well written, and the supplementary results are impressive. However, the paper makes key assumptions which should be clearly articulated. The paper also fails to reference several highly-cited and related topics and as a result makes inaccurate claims.

Reply for general comments: We express our gratitude to the reviewer for thoroughly examining the manuscript and providing valuable feedback and constructive critiques. We have carefully considered their comments to enhance and clarify the manuscript, particularly in presenting the key assumptions related to the proposed displacement measurement system and improving the incorporation of relevant literature references. Please find below a detailed point-by-point response to all comments (reviewers' comments in black, and our replies in blue).

The following pieces of research on using drones for displacement measurement have not been cited. This is just a sample list, there are other papers too that have been missed.

- Bolognini, Michele, et al. "Vision-based modal analysis of built environment structures with multiple drones." *Automation in Construction* 143 (2022): 104550.
- Hoskere, Vedhus, et al. "Vision-based modal survey of civil infrastructure using unmanned aerial vehicles." *Journal of Structural Engineering* 145.7 (2019): 04019062.
- Weng, Yufeng, et al. "Homography - based structural displacement measurement for large structures using unmanned aerial vehicles." *Computer - Aided Civil and Infrastructure Engineering* 36.9 (2021): 1114-1128.

[Response] According to the reviewer suggestions, we added these recent drone-based displacement measurement paper in the revised manuscript (Refs. [35] [29] [31]).

There has also been a large amount of research on phase-based displacement measurement and modal analysis that has not been cited. E.g.,

- Yang, Yongchao, et al. "Blind identification of full-field vibration modes from video measurements with phase-based video motion magnification." *Mechanical Systems and Signal Processing* 85 (2017): 567-590.

The proposed research should be put into the appropriate context after a more detailed literature review.

[Response] We are grateful for the constructive suggestions of adding references to phase-based displacement measurement, which are critical to this research. We have conducted an extensive survey on this theme and included relevant previous studies in the reference list (Refs. [21-23]).

Comment (1): The statements in the manuscript should be appropriately modified to reflect the results from recently published research. For example, this statement may have to be reworded. “Although some recent investigations have ventured into outdoor verification experiments, their accuracy has been constrained, with reported root mean square errors (RMSE) of approximately 2 mm at shooting distances of roughly 5 m”

Reply for comment (1): Thank you for pointing out the issue regarding the expression of reference survey. We have revised the expressions to ensure the correctness of the literature survey in the manuscript.

[Response] We reworded the following statement in the revised Supplementary Information.

(Before revision) “Although some recent investigations have ventured into outdoor verification experiments, their accuracy has been constrained, with reported root mean square errors (RMSE) of approximately 2 mm at shooting distances of roughly 5 m.”

(After revision) “The survey table indicates that the utilization of drone cameras for structural displacement measurement has become a prominent area of research in recent years, with considerable advancements achieved. However, a significant gap remains between concept-proof demonstration experiments and real-world applications in the previous research.”

Comment (2): Details about how the initial angle correction are applied should be provided – is this applied to each reference marker separately? If so, what algorithm is used to automatically detect the marker? Additionally, how are different rotations of the markers reconciled? Is this only applied on the first frame?

Reply for comment (2): Thank you for pointing out this critical point. We agree this part should be clearly presented in the manuscript content and we made throughout revision in the **Supplementary Fig. 2** to provide sufficient details regarding marker processing. In detail, in the current system, we didn't incorporate a marker detection function; practically, we manually generate the bounding boxes

for each marker and further analysis is based on them for the first frame image. For the latter frames, the center of each marker can be detected automatically through a marker tracking algorithm. Then, the marker angle estimation had been performed for each marker individually based on rank minimization. The rationale behind marker angle estimation is that one marker with 0 degrees deviation over y-axis is assumed to exhibit most simple structures in a math form with minimum rank. The details are added in **Supplementary Note 1**. Since the drone stays in hovering mode, we assume that the marker angles remain mostly unchanged for whole video, and thus angle estimation is performed only on the first image frame. After marker angle estimation, we perform counter-direction angle compensation to eliminate the angle effect, which greatly facilitate further process of phase analysis using sampling moiré method.

[**Response**] We added the details for initial angle correction in **Supplementary Note 1**.

Comment (3): The algorithm seems to have an assumption that two stationary points must be available for applicability. This should be emphasized in the abstract and introduction. If this is not the case this should be clarified. Will the algorithm work if the reference markers are out of plane with the measurement marker?

Reply for comment (3): As the reviewer pointed out, two fixed points are used in this study to compensate image blurring with sub-pixel accuracy. We have added an explanatory note to clarify this point in the abstract and introduction. A slight out-of-plane displacement of the reference marker relative to the measurement marker does not interfere with the analysis. This is because the drone and the bridge to be measured are more than 30 m apart, and even if an out-of-plane displacement of a few millimeters occurs in the marker attached on the bridge, it will not make any difference in the captured image.

[**Response**] We have added an explanatory note to clarify this point in the abstract and introduction.

Comment (4): How is the proposed method effected by out-of-plane rotation of the drone (i) at the start of the data capture, and (ii) during the data capture? Similarly, how does the in-plane rotation of the gimbal affect the algorithm? Some clarification on this would be helpful.

Reply for comment (4): The frontal view of the bridge during drone hovering is virtually unaffected by in-plane and out-of-plane rotation after image stabilization developed by our approach using a reference line and recent drone camera with gimbal function.

Our experimental results demonstrated that the 4-DoF image stabilization model introduced in this study effectively facilitates the measurement of millimeter-order deflection of real bridge. However, to confirm these effects more precisely, we further investigated the effect of in-plane rotation of the (drone) camera by computer simulation and out-of-plane rotation through laboratory experiments. The following describes the newly verified simulation and experimental findings.

Firstly, regarding (i) the initial in-plane and/or out-of-plane rotation during the start of image capture, our prior investigations (See Ref. 45; Fig. 7) have confirmed that the sampling moiré method utilized in this study remains unaffected by tilts of a few degrees at the start of the data capture.

Secondly, regarding (ii) during the data capture, the following simulation was performed to clarify the impact of image blurring including the in-plane rotation. A simulated bridge image, sized 4000 by 1600 pixels, underwent affine transformations involving translations in both x and y directions, rotation, and scaling to simulate the image blurring of a drone camera. Additionally, a 2% level of random noise was added to each image to account for typical camera random noise. In both x - and y -directional translations, the image underwent a series of parallel shifts ranging from 0 pixels to a 1-pixel gap, with increments and decrements of 0.1 pixels, ultimately returning to its initial position; Regarding image rotation, a clockwise rotation of 0.001 degrees to 0.01 degrees was executed, followed by a counterclockwise rotation of 0.01 degrees, ultimately restoring the image to its original orientation; In the process of image magnification, it underwent scaling operations, initially enlarging from 1x to 1.001x at 1.0001x step, followed by reduction at 1.0001x step, ultimately restoring it to the original size of 1x. The virtual grid pitch of the markers was defined as 100 mm, and the image was analyzed with a sampling pitch of 20 pixels. Following this, the average value within a 30x30 pixel region centered on the central marker was computed.

The analysis results in **Supplementary Figure 4d** reveal that no significant disparity exists in the displacement magnitude between the scenario devoid of camera shake and the scenario where image correction for translation, rotation, and magnification is applied. This simulation confirms that the proposed method, leveraging two reference markers as “a reference line”, achieves measurement precision akin to the conventional method. Additionally, these simulation outcomes illustrate the

capability of the proposed approach to rectify in-plane rotation with an accuracy of 1/100 pixel relative to the sampling moiré method, remaining unaffected by in-plane rotation.

Thirdly, we conducted an additional experimental test to assess the influence of out-of-plane rotation of the drone camera on the analysis accuracy of the proposed method. We examined the effects of two types of out-of-plane rotations: α , representing rotation in the x -axis direction; and β , representing rotation in the y -axis direction, as well as their combined impact.

Supplementary Fig. 4 Simulated results for the image blurring compensation in 4 DoF case.

a. Simulated bridge image before deformation (left) and after deformation (right). Displacement in **b.** the x -direction and **c.** the y -direction when the image is given a translation, rotation, or scaling operation; **d.** Displacement of the center marker with image blurring compensation.

Supplementary Fig. 5 Photograph of the experimental setup to assess the influence of out-of-plane rotation of the drone camera by utilizing a 6-axis moving stage.

Supplementary Figure 5 depicts the photograph of the experimental setup. Here, the camera (Basler, acA4096-40um; monochrome color) with a focal length of 12 mm lens was fixed on a 6-axis moving stage (Thorlabs, NanoMax 6-Axis Flexure Stage MAX681/M; Theoretical resolution of X, Y, and Z are 1.8 nm, 1.8 nm, and 1.2 nm. Theoretical resolution of Roll, Pitch, Yaw are 0.021 μ Rad, 0.021 μ Rad and 0.021 μ Rad) and the rotation of the moving stage was controlled to simulate the out-of-plane rotation of the drone while hovering. In this verification experiment, the distance from the camera to the marker was 1080 mm, and from Mk-A to Mk-B was 1000 mm. The Mk-C marker was fixed to a 6-axis moving stage (Thorlabs, NanoMax 6-Axis Flexure Stage MAX681/M), and the bridge deflection measurement was reproduced by controlling the amount of movement in the y-direction.

Under the specified experimental conditions, the measurement marker underwent a sequence of movements, including a displacement of 0.5 mm in the y-direction, which was maintained for 2

seconds, followed by a further displacement of 1 mm, also held for 2 seconds. Subsequently, the marker was returned to a 0.5 mm displacement for another 2 seconds before being restored to its original zero displacement position. Concurrently, the camera's tilt angle α was continuously adjusted from 0° to 0.1° , synchronized with the onset of the measurement marker's movement. Similarly, other experiments were conducted involving continuous control of the camera's tilt angle β from 0 to 0.1 degrees, and additional experiments in which both α and β were simultaneously adjusted from 0 to 0.1 degrees.

Supplementary Fig. 6 indicates the experimental results of accuracy verification by rotation angle change in the out-of-plane direction. As shown in **Supplementary Figs. 6b** and **6d**, the error is more than 1.9 mm without correction, whereas after correction, the error is reduced to about 0.03 mm. When the tilt angles of α and β , the rotation angles of the x and y axes, are 0.1 degrees, the rotation of the x -axis is slightly uncorrected, and an error of about 0.03 mm remains, as shown in **Supplementary Figs. 6e** and **6g**.

For a drone equipped with a gimbal (3-axis stabilizer), the angular motion change range, as specified in the product documentation, is less than $\pm 0.005^\circ$. This maximum range is one-tenth of that employed in this verification experiment, resulting in a deflection measurement error due to angle changes that are reduced by a factor of one-tenth. This suggests that the impact of out-of-plane rotation on measurement accuracy is negligible. Besides, the angular change in β affects the displacement in the x -direction. It is incredibly insensitive to deflection measurement, which is displacement in the y -direction. As shown in **Supplementary Fig. 6c**, it can be confirmed that even a minute angular change in β does not affect displacement in the y -direction.

This verification experiment confirmed the ability of the proposed method to achieve the sub-millimeter level of accuracy necessary in the field for deflection measurements, while effectively mitigating measurement errors stemming from out-of-plane rotations during drone video capture in hovering mode.

Supplementary Fig. 6 Experimental results of accuracy verification by rotation angle change in the out-of-plane direction. **a** Photograph of the optical system and three moiré markers with a 5 mm pitch was used. The measuring marker (Mk-C) was attached onto an accurate 6-axis moving stage. The distance between the Mk-A and Mk-C, Mk-B and Mk-C was both 500 mm. Before compensation results in the case of **b** $\Delta\alpha = 0.1$ deg, **c** $\Delta\beta = 0.1$ deg, and **d** $\Delta\alpha = \Delta\beta = 0.1$ deg, respectively. After compensation results in the case of **e** $\Delta\alpha = 0.1$ deg, **f** $\Delta\beta = 0.1$ deg, and **g** $\Delta\alpha = \Delta\beta = 0.1$ deg, respectively.

[Response] In the revised version, additional simulation and validation experiments quantitatively investigate in-plane and out-of-plane rotation effects, presented in **Supplementary Figs. 4, 5, and 6**, respectively.

Reviewer #3 (Remarks to the Authors)

General comments: In the manuscript, the authors proposed a drone-based displacement measurement method. The obtained results should be significant and have a good potential for promoting the application of the traditional sample moiré methods. However, before the manuscript can be accepted for publication, the authors should address the following item.

Reply for general comments: The authors are grateful for the supportive comments and insightful questions raised by the reviewer. We have tried to clarify the point raised by the reviewer and reflected them in the revised manuscript and supplementary information.

Comment (1): In the sampling moiré method, the obtained phase information is relative value which requires manually specifying the zero point of the phase. However, marker C moves with the bridge as a whole and there is no zero point of displacement(phase) in this area. How to choose the zero point of the phase?

Reply for comment (1): We apologize for any confusion caused by the reviewer's understanding due to insufficient explanation. In the sampling moiré method, the phase distribution information obtained is the phase value of the grating, including the initial phase at each position of the grating marker. Images are taken before and after the deformation, and that point's displacement is measured from the phase change (phase difference) of each grating image pixel-by-pixel. In the case of this study, two reference points, Marker-A and Marker-B, are utilized to calculate the relative displacement of Marker-C after image stabilization. Similarly, the amount of displacement of Marker-C is calculated from the images before deformation (before the test vehicle passes through, usually the first frame image) and after deformation (during the passage of the test vehicle, from the second frame image onward) to calculate the time series deflection value caused by the passage of the test vehicle. In this case, the initial phase of the central marker in the first frame image is used as the phase before deformation, the phase after deformation is calculated from the images after the second frame of the same evaluation area, and the displacement of the measurement marker is determined from the phase difference using Equation (3.4). Therefore, this method does not need to find the zero point of the phase.

[Response] In response to this question, we added calculation formula symbols in orange to the illustration in **Supplementary Fig. 3**, which expounds the principles of the sampling moiré (SM) method, for improved clarity. We also emphasized that the detection of zero phase point is unnecessary for SM method to obtain the displacement value.

Comment (2): In the part of “Methods, Coarse-compensation with pixel level precision”, similarity transform is implemented to shift the markers in the video sequence to be within half the marker pitch distance from the marker template in the initial frame. The step is crucial since the subsequent sampling moiré method utilizes relative phase which means any displacement greater than half the marker pitch distance may cause multiplicity. However, how can the authors guarantee the markers will be shifted within the distance in half marker pitch (from the marker template in the initial frame)?

Reply for comment (2): As the reviewer mentioned, image displacement that exceeds more than a half period may cause the sampling moiré method to calculate a displacement that is several times off. For this reason, this study first corrects for image blurring with pixel precision using similarity transform. The details are presented in **Supplementary Note 1**. Through the process, the marker images in each frame of the UAV video are aligned with the first frame. Furthermore, the compensation ensures that the pixel distance between the marker center coordinates remains below half of the marker pitch. In addition to explaining the model, we show experimental results in **Supplementary Fig. 11c**, the displacement of all reference markers of Mk-A and Mk-B after image blurring correction is within about 1 pixel of the top, bottom, left, and right. The image is corrected for the shaking of the grating in this experiment. Since the grating pitch was 13 pixels in the images taken in this experiment, it can be confirmed that it is within 6.5 pixels, which is half the period in all frames. In the unlikely event of a misalignment of more than 7 pixels on the graph in **Supplementary Fig. 11c**, we can identify a problem with the detection of the center coordinates of the reference marker.

[Response] In “**Methods, Coarse-compensation with pixel-level precision**”, we introduced an explanation regarding the process of a pixel-accuracy similarity transform, ensuring that the reference markers shift within half the marker pitch. Besides, the technical details are summarized in **Supplementary Note 1**.

Comment (3): How can the authors fabricate the maker on the bridge structure? Is there any influence from the deviation of the specimen maker from the desired position on the measurement results?

Reply for comment (3): The reference and measurement markers used in the field experiments are the same: a grid pattern with a grid pitch of 200 mm printed on a 3 mm thick plastic plate. These markers (plates) are firmly attached to the surface of the concrete bridge with double-sided adhesive tape (3M sign & display structural joint tape, T410). When attaching the markers, we basically stick

Editorial Note: Parts of Supplementary Figure. 8b below have been redacted as indicated to remove third-party material where no permission to publish could be obtained.

them directly to the location where we want to measure. Typically, the marker is attached after marking the measurement point with white chalk. Misalignment of a few millimeters is almost negligible for bridges ranging from 30 to 100 meters in scale.

Supplementary Figure. 8 Installation of markers for bridge deflection measurement experiment. a photograph of markers designed and fabricated, b photograph of double-sided adhesive tape for attaching to concrete bridge and c fixing point for tape on the back of the marker.

Supplementary Fig. 9 Photograph of large marker attachment in field experiment. To enhance portability, four individual markers with a 200 mm pitch, each measuring 0.5 square meters, are merged. For comparison, a smaller marker employed in measuring the deflection of a 35-meter-long bridge is presented on the right.

[Response] We have added the explanation of the marker specifications and installation in more detail in Supplementary Figs. 8 and 9.

REVIEWERS' COMMENTS

Reviewer #1 (Remarks to the Author):

I am happy that the reviewers have addressed my comments and paper has been improved, and have no further comments.

Reviewer #3 (Remarks to the Author):

The authors already corrected the manuscript according to my comment. I think it can be accepted for publication at the present form.

Response to Reviewer's Comments

Reviewer #1 (Remarks to the Authors)

I am happy that the authors have addressed my comments and paper has been improved, and have no further comments.

Our response: We express sincere gratitude for your publication recommendation and valuable comments that significantly enhance the manuscript of this work.

Reviewer #3 (Remarks to the Authors)

The authors already corrected the manuscript according to my comment. I think it can be accepted for publication at the present form.

Our response: We express sincere gratitude for your publication recommendation and valuable comments that significantly enhance the manuscript of this work.